# Effect of obstructive sleep apnea on cerebrovascular compliance and cerebral small vessel disease

Woo-Jin Lee[1], Keun-Hwa Jung[1], Hyun-Woo Nam[2]*, Yong-Seok Lee[2]*

1 Department of Neurology, Seoul National University Hospital, Seoul, South Korea, 2 Department of Neurology, College of Medicine Seoul National University, Seoul National University-Seoul Metropolitan Government Boramae Medical Center, Seoul, Republic of Korea

* mercades@snu.ac.kr (YSL); hwnam85@gmail.com (HWN)

**Data Availability Statement:** The datasets generated during and/or analysed during the current study are available in the Supplemental Dataset.

## Abstract

Reduced cerebrovascular compliance is the major mechanism of cerebral small vessel disease (SVD). Obstructive sleep apnea (OSA) also promotes SVD development, but the underlying mechanism was not elucidated. We investigated the association among OSA, cerebrovascular compliance, and SVD parameters. This study retrospectively included individuals $\geq$ 50 years of age, underwent overnight polysomnographic (PSG) for the evaluation of OSA, and performed MRI and transcranial Doppler (TCD) within 12 months of interval without a neurological event between the evaluations. TCD parameters for the cerebrovascular compliance included middle cerebral artery pulsatility index (MCA PI) and mean MCA resistance index ratio (MRIR). SVD parameters included white matter hyperintensity (WMH) volume, number of lacunes, enlarged perivascular space (ePVS) score, and the presence of microbleeds or lacunes. Ninety-seven individuals (60.8% male, mean age 70.0±10.5 years) were included. MRIR was associated with higher respiratory distress index (B = 0.003; 95% confidence interval [CI] 0.001–0.005; P = 0.021), while MCA PI was not associated with any of the PSG markers for OSA severity. Apnea-hypopnea index was associated with the log-transformed total WMH volume (B = 0.008; 95% confidence interval [CI] 0.001–0.016; P = 0.020), subcortical WMH volume (B = 0.015; 95% CI 0.007–0.022; P<0.001), total ePVS score (B = 0.024; 95% CI 0.003–0.045; P = 0.026), and centrum semiovale ePVS score (B = 0.026; 95% CI 0.004–0.048; P = 0.019), and oxygen-desaturation index with periventricular WMH volume, independently from age, MCA PI, and MRIR. This study concluded that OSA is associated with reduced cerebrovascular compliance and also with SVD independently from cerebrovascular compliance. Underlying pathomechanistic link might be region specific.

## Introduction

Cerebral small vessel disease (SVD) is a very prevalent age-related phenomenon of the brain [1, 2]. There is no approved treatment for SVD, although a recent randomized study has

**Funding:** This work was supported by a research grant funded by the Korean Society of Neurosonology, Republic of Korea. The funders had no role in study design, data collection and analysis, decision to publish, or preparation of the manuscript.

**Competing interests:** The authors have declared that no competing interests exist.

**Abbreviations:** AHI, apnea-hypopnea index; MCA, middle cerebral artery; MRIR, middle cerebral artery resistance index ratio; ODI, oxygen desaturation index; PI, pulsatility index; RDI, respiratory distress index; RERA, respiratory effort-related arousal; RI, resistance index; SVD, small vessel disease; TCD, Transcranial Doppler sonography.

shown that remote ischemic conditioning reduces the recurrence of ischemic stroke and is an effective therapy for patients with SVD [3]. Reduced compliance in the cerebral arterioles resulting from chronic dysregulated vascular remodeling has been regarded the fundamental pathomechanism of SVD progression, which provokes the blood-brain-barrier (BBB) damage, intermittent hypoperfusion, and impaired clearance of the cerebral waste and ultimately leads to the chronic inflammation and subclinical ischemia in the brain parenchyma [2, 4–7].

Obstructive sleep apnea (OSA) is a very prevalent sleep related breathing disorder in elderly population. The reported incidence of OSA is from 5.6% to 60% in people over 65 while SVD presents an increased one from 6% at age 60 to 28% at age 80 [8]. Aging is associated with an increased risk and severity of OSA, and OSA has also been shown to be related to olfactory disorders, the origin of which could be attributed to disorders of the central nervous system [9].

OSA and SVD share common risk factors such as increasing age, hypertension, diabetes, and obesity, and OSA is associated with the progression of SVD [10–12]. OSA is implicated in the endothelial dysfunction and decreased vascular compliance [7, 9, 13, 14]. A recent study suggested that endothelial dysfunction in OSA may be related to the consequent generation of reactive oxygen species and pro-inflammatory molecules, resulting in microvascular damage in OSA patients [15]. However, the effect of OSA on SVD progression is independent from those cerebrovascular risk factors and its underlying mechanisms might be distinct from the dysregulated vascular remodeling [16, 17]. The candidate pathophysiologic link between OSA and SVD includes abrupt increment in the intrathoracic pressure that interferes with the adequate venous return and cardiac output, frequent arousal, intermittent brain hypoxia, and provocation of arrhythmia [13, 18–20]. Frequent arousal and intermittent hypoxia can recurrently activate the sympathetic nervous system, provoke oxidative stress, and induce parenchymal inflammation [13, 18, 21–23]. Also, altered intrathoracic pressure and frequent arousal might impair the activation of cerebral waste clearance system during sleep [13, 19, 20, 24–26]. Although highly interrelated, discrimination of the major mechanism underlying the progression of SVD in patients with OSA might be important to predict how the adequate treatment of OSA could also modify the progression of SVD [27, 28].

Transcranial Doppler sonography (TCD) is a widely used noninvasive method to measure the blood flow in the brain [29]. Some parameters of TCD are useful indicators for the cerebrovascular compliance [30]. Pulsatility index (PI) evaluates the stiffness of the vessel and also the resistance of the distal arterial bed [31, 32]. Recently, it is reported that a reduction of resistance index (RI) along the middle cerebral artery (MCA RI ratio, MRIR) might be an indirect marker for the compliance of the cerebral small vessels [30].

In this study, we aimed to determine the pathomechanism linking OSA, altered cerebrovascular compliance, and SVD development. In this regard, we investigated whether the polysomnographic (PSG) parameters for the OSA severity was associated with the TCD parameters for the cerebrovascular compliance. Then, we evaluated the associations of the OSA severity factors and the vascular compliance markers with the parameters of cerebral SVD, such as white matter hyperintensity (WMH) volume, enlarged perivascular space (ePVS) score, and the presence of cerebral microbleeds (CMB) or lacunes.

## Materials and methods

### Study subjects and clinical data

The design of this study was approved by the institutional review board of Seoul National University Boramae Hospital. Informed consent was waived by the IRB, as this study did not intervene with the evaluation or treatment process of the patients, and the patients were anonymized during the study process. This retrospective cohort study initially included all

consecutive individuals who were $\geq$ 50 years of age, visited Seoul National University Boramae Hospital between March 2008 and August 2019, underwent overnight PSG evaluation to evaluate for the presumptive diagnosis of OSA, and performed MRI and TCD within 12 months of interval. Among the initially included 175 individuals, the final study population was defined according to the following criteria: (1) no significant ($\geq$ 50%) stenosis in the internal carotid arteries (ICAs) or the MCAs at the initial magnetic resonance angiography (MRA) or TCD; (2) no history of stroke other than an old lacunar stroke ($>$ 90 days); (3) able to carry out daily activities independently without a neurological deficit; and (4) adequate temporal window for a TCD evaluation. According to the criteria, 15 patients with significant stenosis in the ICAs or the MCAs, 45 with a stroke history other than an old lacunar stroke, 11 with fixed neurological deficit or incapable of independent daily living, and 7 with inadequate temporal window were sequentially excluded and the remaining 97 individuals were included for the study analysis. Clinical profiles including demographics, history of lacunar infarction, and presence of hypertension, diabetes mellitus, hyperlipidemia, congestive heart failure, and atrial fibrillation were obtained from the patients' medical record. The design of this study was approved by the institutional review board of Seoul National University Boramae Hospital.

## Transcranial Doppler evaluation

Intracranial arteries were sonographically evaluated using a 2-MHz pulsed-wave and range-gated TCD probe (Digital PMD 100 or ST3 Digital PMD 150; Spencer Technologies; Redmond, WA, USA) with a transmission power level of 100 mW/cm$^2$, pulse repetition frequency of 8000 Hz, filter frequency of 200 Hz, gain of 4 decibels, and range of 200 decibels. The TCD evaluation protocol was standardized for every patient and was conducted by two skilled sonographers with 17 and 10 years of experience, respectively. The flow in the MCA, the anterior cerebral artery, and the internal carotid artery (ICA) was evaluated to exclude stenosis of the anterior cerebral arteries. Peak systolic, minimal diastolic, and mean flow velocities (PSV, MDV, and MFV, cm/s), pulsatility index (PI), and resistance index (RI) were obtained at five points along the M1 portion of the MCA with 5mm intervals insonation depths from the temporal windows [29, 33]. MRIR was designated as the mean ratio of RI along M1 (RI at the shallowest [the most distal] point/ RI at the deepest [the most proximal] point) in each hemisphere [30]. The mean of PI in both MCAs was defined as the average of PIs btained at five insonation points in each hemisphere [25, 30].

## Polysomnography

An overnight video polysomnography (Comet digital recording polysomnographic system, Grass Technologies, West Warwick, RI, USA) were performed in every patient, using 6 electroencephalography leads (C3, C4, F3, F4, O1, and O2 in the international 10–20 system), submental electromyography lead, bilateral electrooculography leads [34]. In order to evaluate the respiratory events, nasal thermal sensor, nasal airflow pressure transducer, thoracic and abdominal strain gauges, microphone, position sensor, and finger pulse oximetry were included in the PSG monitoring devices [34].

Sleep stages, arousals, and respiratory events including apnea, hypopnea, and respiratory effort-related arousal (RERA) were scored according to the 2017 version of the American Academy of Sleep Medicine guidelines [34]. Apnea-hypopnea index (AHI) was defined as the mean number of apnea or hypopnea events per hour, respiratory-distress index (RDI) as the mean number of apnea, hypopnea, or RERA events per hour, oxygen desaturation index (ODI) as the mean number of drop in oxygen saturation $\geq$3% in comparison with

immediately preceding basal value, and arousal index as the mean number of arousals per hour [35, 36]. A significant OSA (moderate to severe degree) was defined as an AHI value of ≥15/h [34].

## Magnetic resonance imaging analysis

MRI was performed using 1.5-T units (Intera Achieva; Philips Medical Systems, Best, the Netherlands, Signa Excite; GE Medical Systems, Milwaukee, WI, USA, and Verio; Siemens Medical Solutions, Erlangen, Germany) with the protocols including T1/T2-weighted images, fluid-attenuated inversion recovery (FLAIR) sequences, gradient recalled echo/susceptibility weight images (GRE/SWI), and time-of-flight magnetic resonance angiography (MRA). T1/T2-weighted images and FLAIR was obtained with the parameters as follows: slice number = 24–30, slice thickness/gap = 4.0–5.0/0.0–1.0 mm, repetition time/echo time (TR/TE) = 9000–9900/97–163 milliseconds, field-of-view = 220–256 × 220–256 mm, and matrix = 320–352 × 192–256. A neurologist (WJL, with 9 years of experience and blinded to other data) reviews the images to exclude preexisting territorial stroke lesions and the MRA to exclude significant stenosis in the cerebral arteries.

The volumetric analysis of WMH was performed by a neurologist (WJL) blinded to other data, according to the protocols previously described. In brief, FLAIR sequences were registered into an offline workstation. A freeware NeuRoi (Nottingham university, Nottingham, UK) was used to identify the WMH lesion, designate its boundaries, calculate the volume [25, 30, 37]. The periventricular and subcortical WMH volume was separately measured and normalized by the total intracranial volume [30, 38]. The reproducibility of the volumetric analysis was established as good in the previous report [30].

ePVS was designated as small and sharply demarcated ovoid or linear shaped lesions which appeared as hyperintensity in T2 weighted images and hypointensity in T1 weighted images. ePVS was rated in basal ganglia (BG) and centrum semiovale (CS) using the Wardlaw scale of which score range is 0–4 (0 for no, 1 for 1–10, 2 for 11–20, 3 for 21–40, and 4 for >40 ePVS) and total score range is 0–8 [7, 39]. CMB was defined as a 2–10 millimeter punctate hypointensity in GRE/SWI images and lacune was designated as a hyperintense lesion with central hypointensity in FLAIR images [7].

## Statistical analysis

SPSS 25.0 (SPSS Inc., Chicago, IL, USA) was used for all statistical analyses. Pearson coefficients was used to measure correlations between continuous variables, Spearman's Rho to measure correlations between ordinal variables, paired *t* test to compare mean values, and chi-square test to compare the frequencies. Age, sex, and variables with *P* values <0.15 in univariate analyses were included in regression analyses using backward elimination. Linear regression analyses were performed to evaluate the factors associated with WMH volume parameters. WMH volumes were log-transformed to obtain a normal distribution. Multi-collinearity between variables was estimated using the variance inflation factor (VIF). A scatterplot of the standardized predicted values and the standardized residuals was used to check for the linearity of the regression model. Ordinal regression analyses were performed to evaluate the factors associated with ePVS scores. For all analyses, *P* values < 0.05 were considered as statistically significant.

## Results

Ninety-seven patients, including 59 males (60.8%) and 38 females (39.2%), with a mean age of 70.0±10.5 years (range, 50–88 years) were included. The median interval between the

evaluations were 82 days (interquartile range, IQR 49–141 days, range -186–236 days) for the PSG and the MRI, 62 days (IQR 28–123 days, range -283–361 days) for the PSG and the TCD, and 17 days (IQR 10–23 days, range -356–72 days) for the MRI and TCD. The indications for the MRI and the TCD analysis were for follow-up evaluation of a pre-existing cerebral WMH in 16 (16.5%) patients or of an old lacunar infarction in 9 (9.3%) patients, and investigations for nonspecific neurologic symptoms such as headache, dizziness, or subjective memory impairment in 72 (74.2%) patients.

In the PSG evaluation, the mean value of total sleep time was 322.6±64.1 min, sleep efficiency was 77.2±14.0%, AHI was 19.1±18.2 /h, RDI was 20.0±18.2 /h, ODI was 20.2±18.3 /h, and arousal index was 25.0±15.0 /h. In the TCD evaluation, the mean MCA MFV was 52.3 ±16.6 cm/sec, MCA PI was 0.80±0.14, and MRIR was 0.99±0.05. The mean total WMH volume was 0.50±0.41%, subcortical WMH volume was 0.26±0.22%, and periventricular WMH volume was 0.23±0.20% of the total intracranial volume. The median total ePVS score was 4 (2–5), CS ePVS score was 2 (1–3), and BG ePVS score was 1 (1–2). CMB was present in 9 (9.3%) patients and lacunes in 11 (11.3%). Compared to the group with AHI < 15 (N = 48), the group with AHI ≥ 15 (N = 49) showed a higher frequency of male sex, hypertension, hyperlipidemia, higher proportion of N1 sleep stage, lower proportion of N3 sleep stage, higher RDI, ODI, and arousal index in PSG, and higher subcortical WMH volume. However, TCD parameters and the other SVD parameters were comparable between the groups with or without a significant OSA (**Table 1**).

In the univariate analyses for the MCA PI, higher age, shorter total sleep time, lower sleep efficiency, and higher MCA MFV were associated with higher MCA PI, while none of the OSA parameters or other clinical parameters showed a significant association with MCA PI (**S1** and **S2 Tables**). Following multivariate linear regression analysis revealed that higher age was the only parameter significantly associated with MCA PI (B coefficient = 0.007; 95% confidence interval [CI] 0.004–0.009; $P$<0.001, **Table 2**).

For MRIR, univariate analyses showed that RDI, the proportion of N3 sleep stage, and the presence of atrial fibrillation were associated with MRIR (**S1** and **S2 Tables**). In linear regression analysis, higher RDI (B = 0.003; 95% CI 0.001–0.005; $P$ = 0.021) and lower proportion of N3 sleep stage (B = -0.001; 95% CI -0.002–0.000; $P$ = 0.030) were associated with higher MRIR (**Table 2**).

Univariate analyses for the total WMH volume revealed that higher age, shorter total sleep time, higher AHI, RDI, ODI, and arousal index, higher MCA PI, higher MRIR, and the presence of diabetes mellitus or hyperlipidemia were associated with higher WMH volume (**S1** and **S2 Tables**). In following multivariate linear regression analysis, age (B = 0.032; 95% CI 0.018–0.046; $P$<0.001), AHI (B = 0.008; 95% CI 0.001–0.016; $P$ = 0.020), MCA PI (B = 1.001; 95% CI 0.011–1.990; $P$ = 0.047), and MRIR (B = 3.775; 95% CI 1.232–6.318; $P$ = 0.004) were associated with the log-transformed total WMH volume (**Table 3**). For subcortical WMH volume, age (B = 0.026; 95% CI 0.013–0.039; $P$<0.001), AHI (B = 0.015; 95% CI 0.007–0.022; $P$<0.001), and MRIR (B = 3.964; 95% CI 1.346–6.583; $P$ = 0.013) were associated with the log-transformed subcortical WMH volume. However, the log-transformed periventricular WMH volume was associated with ODI (B = 0.006; 95% CI 0.001–0.010; $P$ = 0.049), along with age (B = 0.032; 95% CI 0.018–0.046; $P$<0.001), MCA PI (B = 1.107; 95% CI 0.005–2.210; $P$ = 0.049) and MRIR (B = 3.512; 95% CI 1.220–5.803; $P$ = 0.016) (**Tables 3** and **S1** and **S2** for the univariate analyses). In every analysis, the scatterplot of the standardized predicted values and the standardized residuals showed a random and even distribution of the standardized residuals around the zero line. VIF values for each variable were <1.7.

In univariate analyses for ePVS, total ePVS score was associated with higher age, AHI, RDI, MRIR, and MCA PI, and the presence of diabetes mellitus (**S1** and **S2 Tables**). Following

**Table 1. Clinical, laboratory, and white matter hyperintensity profiles of the study population.**

| | Total (N = 97) | AHI < 15 (N = 48) | AHI ≥ 15 (N = 49) | P |
|---|---|---|---|---|
| Age (years) | 70.0±10.5 | 69.2±10.4 | 70.9±10.6 | 0.250 |
| Male sex (%) | 59 (60.8) | 19 (39.6) | 40 (81.6) | <0.001** |
| Hypertension (%) | 60 (61.9) | 24 (50.0) | 36 (73.5) | 0.017* |
| Diabetes mellitus (%) | 29 (29.9) | 11 (22.9) | 18 (36.7) | 0.140 |
| Hyperlipidemia (%) | 29 (29.9) | 9 (18.8) | 20 (40.8) | 0.017* |
| Heart failure (%) | 3 (3.1) | 2 (4.2) | 1 (2.0) | 0.550 |
| Atrial fibrillation (%) | 7 (7.2) | 3 (6.3) | 4 (8.2) | 0.719 |
| **Polysomnography findings** | | | | |
| Time in bed (min) | 420.3±44.9 | 422.9±33.7 | 417.9±54 | 0.585 |
| Total sleep time (min) | 322.6±64.1 | 326.3±64.1 | 318.9±64.6 | 0.569 |
| Sleep efficiency (%) | 77.2±14.0 | 77.8±13.7 | 76.6±14.5 | 0.689 |
| Stage N1 (%) | 21.4±13.5 | 15.1±10.5 | 27.6±13.4 | <0.001** |
| Stage N2 (%) | 53.0±60.4 | 59.8±74.4 | 46.6±42.9 | 0.288 |
| Stage N3 (%) | 15.3±9.6 | 17.4±10.8 | 13.3±7.8 | 0.034 |
| Stage REM sleep (%) | 18.8±8.5 | 18.9±7.0 | 18.6±9.8 | 0.875 |
| Sleep-onset latency (min) | 12.2±12.9 | 13.8±14.6 | 10.5±10.9 | 0.216 |
| REM sleep latency (min) | 109.5±72.4 | 102.8±51.3 | 116.2±88.4 | 0.362 |
| Apnea-hypopnea index (/h) | 19.1±18.2 | 4.8±4.2 | 33.1±15.6 | <0.001** |
| Respiratory distress index (/h) | 20.0±18.2 | 6.0±5.3 | 33.7±15.7 | <0.001** |
| Oxygen-desaturation index (/h) | 20.2±18.3 | 5.1±4.2 | 34.5±15.7 | <0.001** |
| Arousal index (/h) | 25.0±15.0 | 17.0±9.0 | 32.9±15.6 | <0.001** |
| **Transcranial Doppler and MRI findings** | | | | |
| MCA MFV (cm/sec) | 52.3±16.6 | 50.8±11.0 | 53.8±20.7 | 0.372 |
| MCA PI | 0.80±0.14 | 0.78±0.13 | 0.82±0.15 | 0.163 |
| MRIR | 0.99±0.05 | 0.99±0.05 | 0.99±0.05 | 0.336 |
| WMH total volume (%) | 0.50±0.41 | 0.44±0.34 | 0.56±0.47 | 0.137 |
| WMH subcortical volume (%) | 0.26±0.22 | 0.22±0.17 | 0.31±0.25 | 0.045* |
| WMH periventricular volume (%) | 0.23±0.20 | 0.22±0.17 | 0.25±0.22 | 0.363 |
| Total ePVS score | 4 (2–5) | 4 (2–4) | 4 (2.5–5) | 0.107 |
| CS ePVS score | 2 (1–3) | 2 (1–3) | 2 (2–3) | 0.113 |
| BG ePVS score | 1 (1–2) | 1 (1–2) | 1 (1–2) | 0.225 |
| Presence of CMB | 9 (9.3) | 5 (10.4) | 4 (8.2) | 0.706 |
| Presence of Lacunes | 11 (11.3) | 5 (10.4) | 6 (12.2) | 0.779 |

Data are reported as number (percentage), as mean± standard deviation, or as median (interquartile range, IQR). AHI: apnea-hypopnea index, REM: rapid eye movement, MCA: middle cerebral artery, MFV: mean flow velocity, PI: pulsatility index, MRIR: mean middle cerebral artery resistance index ratio, WMH: white matter hyperintensity, ePVS: enlarged perivascular space, CS: centrum semiovale, BG: basal ganglia, and CMB: cerebral microbleeds.

*P<0.05 and

**P<0.01.

multivariate ordinal regression analysis revealed that total ePVS score was associated with age (B = 0.069; 95% CI 0.001–0.016; P = 0.001), AHI (B = 0.024; 95% CI 0.003–0.045; P = 0.026), and MRIR (B = 10.264; 95% CI 3.183–17.346; P = 0.020, **Table 4**). Similarly, CS ePVS score was associated with age (B = 0.061; 95% CI 0.020–0.103; P = 0.004), AHI (B = 0.026; 95% CI 0.004–0.048; P = 0.019), and MRIR (B = 7.264; 95% CI 2.183–12.345; P = 0.036). However, BG ePVS was associated with age (B = 0.062; 95% CI 0.016–0.108; P = 0.008), MRIR (B = 9.764; 95% CI 2.967–16.373; P = 0.031), and the presence of hypertension (B = 1.602; 95% CI 0.652–

**Table 2. Linear regression analyses for factors associated with the cerebrovascular compliance markers.**

| MCA PI[a] | B (95% CI) | β | P |
|---|---|---|---|
| | 0.343 (0.169–0.516) | | <0.001** |
| Age (year) | 0.007 (0.004–0.009) | 0.480 | <0.001** |
| **MRIR**[b] | B (95% CI) | β | P |
| | 1.002 (0.984–1.020) | | <0.001** |
| Stage N3 (%) | -0.001 (-0.002–0.000) | -0.210 | 0.030* |
| Respiratory distress index (/h) | 0.003 (0.001–0.005) | 0.226 | 0.021* |

[a]$R^2$ = 0.230 and $P$<0.001 and

[b]$R^2$ = 0.247 and $P$<0.001 for the linear regression equation.

B: unstandardized coefficient, β: standardized coefficient, CI: confidence interval, MCA: middle cerebral artery, PI: pulsatility index, and MRIR: mean middle cerebral artery resistance index ratio.

*$P$<0.05 and

**$P$<0.01.

2.551; $P$ = 0.001), but only marginally with RDI (B = 0.022; 95% CI -0.001–0.045; $P$ = 0.060, **Tables 4** and **S1** and **S2** for the univariate analyses). None of the clinical, PSG, or TCD parameters was associated with the presence of CMB or lacunes. Multivariate analysis was not performed due to the small numbers of the patients with CMB or lacunes (**S3 Table**).

**Table 3. Linear regression analyses for factors associated with the log-transformed value of the white-matter hyperintensity volumes.**

| WMH total volume (%, log-transformed)[a] | B (95% CI) | β | P |
|---|---|---|---|
| Intercept | -8.135 (-10.815–-5.456) | | <0.001** |
| Age (year) | 0.032 (0.018–0.046) | 0.415 | <0.001** |
| Apnea-hypopnea index (/h) | 0.008 (0.001–0.016) | 0.191 | 0.020* |
| MCA PI | 1.001 (0.011–1.990) | 0.180 | 0.047* |
| MRIR | 3.775 (1.232–6.318) | 0.234 | 0.004** |
| **WMH subcortical volume** (%, log-transformed)[b] | B (95% CI) | β | P |
| Intercept | -7.977 (-10.528–-5.427) | 0.000 | <0.001** |
| Age (year) | 0.026 (0.013–0.039) | 0.356 | <0.001** |
| Apnea-hypopnea index (/h) | 0.015 (0.007–0.022) | 0.336 | <0.001** |
| MCA PI | 1.002 (-0.017–2.021) | 0.178 | 0.054 |
| MRIR | 3.964 (1.346–6.583) | 0.243 | 0.013* |
| **WMH periventricular volume** (%, log-transformed)[c] | B (95% CI) | β | P |
| Intercept | -9.217 (-12.04–-6.394) | 0.000 | <0.001** |
| Age (year) | 0.032 (0.018–0.046) | 0.418 | <0.001** |
| Oxygen desaturation index (/h) | 0.006 (0.001–0.010) | 0.192 | 0.046* |
| MCA PI | 1.107 (0.005–2.210) | 0.188 | 0.049* |
| MRIR | 3.512 (1.220–5.803) | 0.224 | 0.016** |

[a]$R^2$ = 0.424 and $P$<0.001,

[b]$R^2$ = 0.676 and $P$<0.001, and

[c]$R^2$ = 0.619 and $P$<0.001 for the linear regression equation.

B: unstandardized coefficient, β: standardized coefficient, CI: confidence interval, WMH: white matter hyperintensity, MCA: middle cerebral artery, PI: pulsatility index, and MRIR: mean middle cerebral artery resistance index ratio.

*$P$<0.05 and

**$P$<0.01.

**Table 4. Linear regression analyses for factors associated with the enlarged perivascular space scores.**

| Total ePVS score | B (95% CI) | P |
|---|---|---|
| Age (year) | 0.069 (0.029–0.108) | 0.001** |
| Apnea-hypopnea index (/h) | 0.024 (0.003–0.045) | 0.026* |
| MCA PI | 2.556 (-0.441–5.554) | 0.095 |
| MRIR | 10.264 (3.183–17.346) | 0.020* |
| **Thresholds** | B (SE) | |
| 0 | 18.388 (4.206) | |
| 1 | 19.487 (4.227) | |
| 2 | 20.846 (4.282) | |
| 3 | 21.631 (4.317) | |
| 4 | 23.128 (4.383) | |
| 5 | 25.197 (4.481) | |
| **CS ePVS score** | B (95% CI) | P |
| Age (year) | 0.061 (0.020–0.103) | 0.004** |
| Apnea-hypopnea index (/h) | 0.026 (0.004–0.048) | 0.019* |
| MCA PI | 2.753 (-0.363–5.868) | 0.083 |
| MRIR | 7.264 (2.183–12.345) | 0.036* |
| **Thresholds** | B (SE) | |
| 0 | 13.385 (4.324) | |
| 1 | 14.946 (4.367) | |
| 2 | 16.833 (4.434) | |
| 3 | 19.793 (4.554) | |
| **BG ePVS score** | B (95% CI) | P |
| Age (year) | 0.062 (0.016–0.108) | 0.008** |
| Respiratory distress index (/h) | 0.022 (-0.001–0.045) | 0.060 |
| MCA PI | 2.137 (-1.183–5.458) | 0.207 |
| MRIR | 9.764 (2.967–16.373) | 0.031* |
| Hypertension | 1.602 (0.652–2.551) | 0.001** |
| **Thresholds** | B (SE) | |
| 0 | 19.491 (4.88) | |
| 1 | 23.079 (5.082) | |
| 2 | 26.434 (5.263) | |

[a]$R^2$ = 0.388 and $P<0.001$, [b]$R^2$ = 0.384 and $P$ = 0.001, and [c]$R^2$ = 0.378 and $P<0.001$ for the regression equation.

B: unstandardized coefficient, CI: confidence interval, ePVS: enlarged perivascular space, CS: centrum semiovale, BG: basal ganglia, MCA: middle cerebral artery, PI: pulsatility index, and MRIR: mean middle cerebral artery resistance index ratio.

*$P<0.05$ and

**$P<0.01$.

## Discussion

The purpose of the current study was to investigate whether the OSA severity was associated with the cerebrovascular compliance and to evaluate the associations of the OSA severity factors with the SVD parameters. This study found that RDI was significantly associated with the TCD marker for cerebral microvascular compliance, MRIR, but not with another marker for cerebrovascular remodeling, MCA PI. AHI and those cerebral vascular remodeling parameters were independently associated with the cerebral WMH volume and ePVS scores. Additionally, AHI was

significantly associated with the subcortical SVD parameters such as subcortical WMH volume and CS ePVS score, whereas ODI was associated with WMH volume in deep structures. These findings are concordant to that of a recent meta-analysis, which reported that moderate to severe OSA is positively associated with WMH and silent brain infarction, but not with CMBs [40].

The major finding of the current study is that OSA is associated with cerebral SVD independently from the reduced cerebrovascular compliance. This finding suggests that OSA influences the pathogenesis of SVD by multiple mechanisms and some of them might be distinct from the chronic cerebrovascular remodeling. Considering that chronic aging related cerebrovascular remodeling process has progressive and irreversible feature [2, 41, 42], the finding of this study also indicates that the effect of OSA on the SVD might be reversible in some extent. Previous studies reported that continuous positive airway pressure (CPAP) treatment reversed the altered diffusion tensor image parameters for the white matter integrity and the voxel-based morphometry based gray matter volume parameters in patients with OSA, and explained the reversibility by the improvement of the early osmotic changes in the cells induced by mild ischemia and the inflammation that impairs the cell membrane integrity [27, 28]. Additionally, surgical treatment for OSA by relocation pharyngoplasty, a variant of uvulo-palatopharyngoplasty after, also improved the level of high sensitivity C-reactive protein and reduced cardiovascular risk in subjects with OSA [43, 44].

Some plausible pathophysiologic link might explain the association between OSA and SVD independent from the vascular remodeling process. Recent findings that sleep activates the perivascular system function, which is a major pathway for the clearance of cerebral wastes to maintain brain homeostasis of which dysfunction is associated with the pathogenesis of SVD [19, 45, 46]. Sleep induced amplification of the very low frequency oscillations in the cerebral flow velocity might mediate the activation of this system, and disturbed sleep is associated with the impaired regulation of the major cerebral wastes such as β-amyloid or tau proteins [19, 25]. Adequate regulation of the cerebral blood flow pulsation and the CSF recirculation is required for the functioning of the cerebral waste clearance system [37, 47]. Therefore, respiratory event related abrupt intrathoracic pressure increment and the disruption of the sleep structures by frequent arousals might significantly interfere with the cerebral waste clearance during sleep [13, 26]. Additionally, intermittent brain hypoxia provokes oxidative stress and activates inflammatory cascade, which results in the disruption of the BBB integrity and the glial and neuronal cell damage [18, 24]. Respiratory event during sleep also provokes paroxysmal cardiac arrhythmia that is associated with the pathogenesis of SVD [20, 48]. As those pathologic effects can be efficiently prevented by CPAP treatment, OSA might be considered as another modifiable risk factor for the cerebral SVD progression, independent from the conventional vascular risk factors (**Fig 1**).

Another major finding of the current study is that the effect of AHI on the SVD was different between the brain regions. AHI was significantly associated with the subcortical SVD parameters. However, the SVD parameters of periventricular or basal ganglia was not associated with AHI but rather with ODI, and the conventional cerebrovascular compliance factors such as the presence of hypertension, MCA PI, and MRIR. This might be due to the difference in the dominant pathomechanism of SVD between the subcortical and deep brain structures. As periventricular and basal ganglia have less production of cerebral wastes and are supplied the highly compliant perforating arterioles ramifying directly and perpendicularly from the M1 of MCA, those regions might be less dependent to the sleep-related cerebral waste clearance system function and more susceptible to the conventional cerebrovascular remodeling or the hypoxic damage resulting from the respiratory events during sleep, measured by ODI [7, 35, 37, 38, 47, 49].

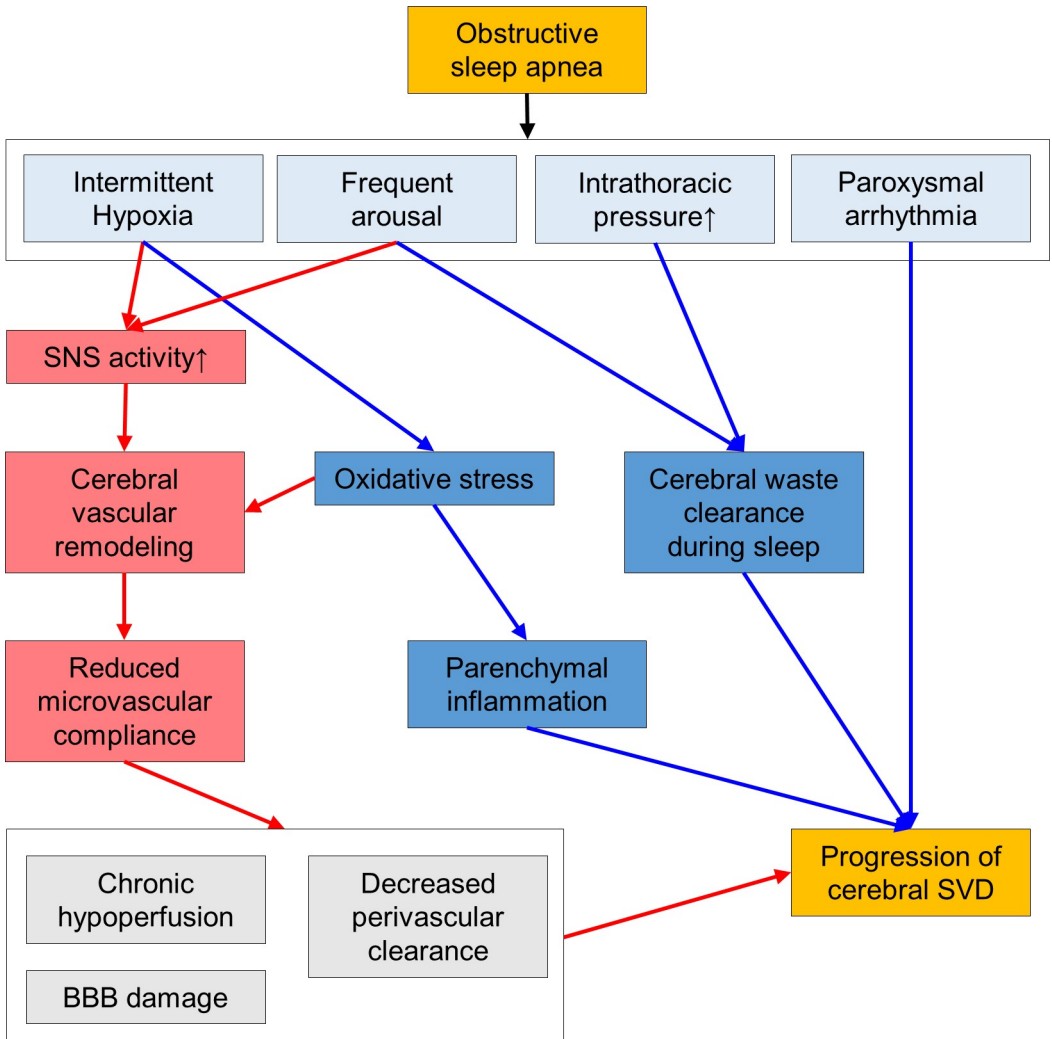

**Fig 1. Schematic explanation of the pathophysiologic link between obstructive sleep apnea and cerebral small vessel disease.** The red lines with arrowheads denotes the pathophysiologic mechanisms dependent on the cerebral vascular remodeling process and the blue lines denotes the mechanisms independent from the cerebral vascular remodeling process. SNS: sympathetic nerve system, BBB: blood brain barrier, and SVD: small vessel disease.

Additionally, this study also observed that RDI was associated with higher MRIR, a marker for a reduced compliance of the perforating arterioles ramifying from M1 of MCA [30]. Considering that RDI reflects the burden of sympathetic activation by sleep related respiratory events, and is associated with the development of hypertension and coronary artery disease [50, 51], this association indicates that sympathetic activation during sleep has a major role in the chronic reduction of the cerebral microvascular compliance.

Several limitations in this study need to be addressed. First, due to the retrospective design, the clinical profiles of the study population and the indications for the MRI and the TCD evaluations were heterogeneous. For instance, a history of lacunar infarction or subjective memory impairment might have implicated in the severity of cerebral WMH. Second, the intervals between the PSG, MRI, and TCD were unstandardized. Therefore, the WMH burden at the time of MRI evaluation might be different from that at the time when PSG or the TCD was

performed, although the patients were stable with neurological status during the study intervals. Third, due to the relatively small study population, some factors with actual pathophysiologic relevance might have not reached a statistical significance. Fourth, due to the cross-sectional design, the findings of the current study did not ensure the causative relationship among the OSA, vascular remodeling, and the progression of WMH. Future prospective studies with larger study populations, standardized follow-up protocols, and evaluation for the effect of CPAP treatment are warranted to clarify the clinical significance of OSA in the pathomechanism of cerebral WMH and the possible therapeutic effect.

## Supporting information

**S1 Table. Correlation coefficients among the continuous or ordinal variables.**
(DOCX)

**S2 Table. Univariate analyses for categorical parameters associated with white-matter hyperintensity volume.**
(DOCX)

**S3 Table. Profiles of clinical, polysomnography, and transcranial cerebrovascular compliance markers in patients with or without cerebral microbleeds or lacunes.**
(DOCX)

**S1 Checklist. STROBE statement—checklist of items that should be included in reports of cross-sectional studies.**
(DOCX)

**S1 Dataset.**
(XLSX)

## Author Contributions

**Conceptualization:** Woo-Jin Lee, Yong-Seok Lee.

**Data curation:** Woo-Jin Lee, Hyun-Woo Nam.

**Formal analysis:** Woo-Jin Lee.

**Funding acquisition:** Woo-Jin Lee.

**Methodology:** Keun-Hwa Jung.

**Project administration:** Yong-Seok Lee.

**Resources:** Keun-Hwa Jung, Hyun-Woo Nam.

**Supervision:** Hyun-Woo Nam, Yong-Seok Lee.

**Validation:** Hyun-Woo Nam, Yong-Seok Lee.

**Writing – original draft:** Woo-Jin Lee.

**Writing – review & editing:** Keun-Hwa Jung, Hyun-Woo Nam, Yong-Seok Lee.

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
