## [Decision Letter · Decision Letter 0]

7 Oct 2021

PONE-D-21-27488Effect of obstructive sleep apnea on cerebrovascular compliance and cerebral small vessel diseasePLOS ONE

Dear Dr. Lee,

Thank you for submitting your manuscript to PLOS ONE. After careful consideration, we invite you to submit a revised version of the manuscript that addresses the points raised during the review process.

We look forward to receiving your revised manuscript.

Kind regards,

Academic Editor

PLOS ONE

“This work was supported by a research grant funded by the Korean Society of Neurosonology, Republic of Korea.”

Please include this amended Role of Funder statement in your cover letter; we will change the online submission form on your behalf

“This work was supported by a research grant funded by the Korean Society of Neurosonology, Republic of Korea.”

We note that you have provided additional information within the Acknowledgements Section. Please note that funding information should not appear in the Acknowledgments section or other areas of your manuscript. We will only publish funding information present in the Funding Statement section of the online submission form.

“This work was supported by a research grant funded by the Korean Society of Neurosonology, Republic of Korea”

7.Please review your reference list to ensure that it is complete and correct. If you have cited papers that have been retracted, please include the rationale for doing so in the manuscript text, or remove these references and replace them with relevant current references. Any changes to the reference list should be mentioned in the rebuttal letter that accompanies your revised manuscript. If you need to cite a retracted article, indicate the article’s retracted status in the References list and also include a citation and full reference for the retraction notice.

Additional Editor Comments:

Well written study. Reviewers suggest minor revisions

Reviewers' comments:

Reviewer's Responses to Questions

**Comments to the Author**

1. Is the manuscript technically sound, and do the data support the conclusions?

Reviewer #1: Yes

Reviewer #2: Yes

2. Has the statistical analysis been performed appropriately and rigorously? 

Reviewer #1: Yes

Reviewer #2: Yes

3. Have the authors made all data underlying the findings in their manuscript fully available?

Reviewer #1: Yes

Reviewer #2: Yes

4. Is the manuscript presented in an intelligible fashion and written in standard English?

Reviewer #1: Yes

Reviewer #2: Yes

5. Review Comments to the Author

Reviewer #1: Interesting paper, minor corrections to improve the quality:

Introduction

-line 55, Subcortical ischemic vascular dementia (SIVD) is very common among older people, but has no approved treatment. A recent randomized study has shown that remote ischemic conditioning (RIC) reduces the recurrence of ischemic stroke and is an effective therapy for patients with SIVD. please cite doi: 10.1186/s12883-019-1435-y.

- line 60, elderly patients are known to have an increased risk of OSAS, with a prevalence of more severe forms after 65 than younger patients. The same severity of obstructive sleep apnea has also been shown to be related to olfactory disorders in the patient with obstructive sleep apnea, the origin of which could be attributed to disorders of the central nervous system. please cite doi:10.1007/s00405-020-06316-w.

line 70, please cite doi:10.14639/0392-100X-895

Methods and results well structured and clearly expressed

Discussion

line 311, also surgical approaches demonstrated improvement of obstructive sleep apnea syndrome (OSAS) and changes in high sensitivity C-reactive protein (hs-CRP) concentrations have been reported after transfer pharyngoplasty (RP), an elevated variant of uvulopalatopharyngoplasty, reducing cardiovascular risk in subjects with obstructive sleep apnea.

please cite doi:10.1177/0194599810395104 and doi:10.1007/s00405-020-05883-2

Reviewer #2: The paper examined an interesting topic. I support its approval suggesting some minor corrections to improve the quality:

Introduction:

Line 60: Reported incidence of OSA is from 5.6% to 60% in people over 65 (cite doi: 10.3390/ijerph17031120) while SVD presents an increased one from 6% at age 60 to 28% at age 80 (cite doi: 10.3390/ijerph17031120).

Line 70: The endothelial dysfunction, may be related to the consequent generation of reactive oxygen species (ROS) and pro-inflammatory molecules, resulting in microvascular damage in OSA patients (cite doi.org/10.3390/jcm10020277).

Methods: clearly expressed and structured

Results: well written and exhaustive.

Discussion:

Line 295: A recent meta-analysis reported that moderate to severe OSA is positively associated with WMH and SBI but not CMBs or PVS (cite doi: 10.1093/sleep/zsz264).

In all paper put the dots after the brackets.

6. PLOS authors have the option to publish the peer review history of their article (what does this mean?). If published, this will include your full peer review and any attached files.

Reviewer #1: No

Reviewer #2: No

---

## [Author Response · Author response to Decision Letter 0]

13 Oct 2021

Effect of obstructive sleep apnea on cerebrovascular compliance and cerebral small vessel disease

Manuscript No.: PONE-D-21-27488

We are grateful for the thorough review and kind advices that were of great help to improve the quality of our manuscript. The manuscript was revised to clarify the issues raised by the reviewers. The changes and authors’ opinions according to the reviewers’ comments are summarized below. We also enabled the "track changes" feature in the manuscript.

REVIEWER #1:

General Comments:

Interesting paper, minor corrections to improve the quality

Specific Comments:

#1. Introduction

-Line 55, Subcortical ischemic vascular dementia (SIVD) is very common among older people, but has no approved treatment. A recent randomized study has shown that remote ischemic conditioning (RIC) reduces the recurrence of ischemic stroke and is an effective therapy for patients with SIVD. please cite doi: 10.1186/s12883-019-1435-y.

Line 60, elderly patients are known to have an increased risk of OSAS, with a prevalence of more severe forms after 65 than younger patients. The same severity of obstructive sleep apnea has also been shown to be related to olfactory disorders in the patient with obstructive sleep apnea, the origin of which could be attributed to disorders of the central nervous system. please cite doi:10.1007/s00405-020-06316-w.

Line 70, please cite doi:10.14639/0392-100X-895

**Response: We thank the reviewer for this discerning advices and recommendation of important up-to-date articles regarding the pathomechanism of OSA. We revised the relevant sentences in the Introduction and added the articles the reviewer has recommended. 

#2. Line 311, also surgical approaches demonstrated improvement of obstructive sleep apnea syndrome (OSAS) and changes in high sensitivity C-reactive protein (hs-CRP) concentrations have been reported after transfer pharyngoplasty (RP), an elevated variant of uvulopalatopharyngoplasty, reducing cardiovascular risk in subjects with obstructive sleep apnea.

please cite doi:10.1177/0194599810395104 and doi:10.1007/s00405-020-05883-2

**Response: We again thank the reviewer for this comment. We revised the Discussion section and added the articles the reviewer has recommended.

REVIEWER #2:

General Comments: 

The paper examined an interesting topic. I support its approval suggesting some minor corrections to improve the quality.

Specific Comments:

1. Introduction:

Line 60: Reported incidence of OSA is from 5.6% to 60% in people over 65 (cite doi: 10.3390/ijerph17031120) while SVD presents an increased one from 6% at age 60 to 28% at age 80 (cite doi: 10.3390/ijerph17031120).

Line 70: The endothelial dysfunction, may be related to the consequent generation of reactive oxygen species (ROS) and pro-inflammatory molecules, resulting in microvascular damage in OSA patients (cite doi.org/10.3390/jcm10020277).

**Response: We thank the reviewer for this comments and recommendation of important up-to-date articles regarding the pathomechanism of OSA. We revised the relevant sentences in the Introduction and added the articles the reviewer has recommended.

2. Line 295: A recent meta-analysis reported that moderate to severe OSA is positively associated with WMH and SBI but not CMBs or PVS (cite doi: 10.1093/sleep/zsz264).

**Response: We thank the reviewer for informing us about this article. According to the reviewer’s recommendation, we revised the relevant sentences and added the articles the reviewer has recommended.

3. In all paper put the dots after the brackets.

**Response: We applied the changes the reviewer has requested, throughout the manuscript. 

Once again, we thank the editor and the reviewers for the thoughtful comments about our work. We hope that this revision will meet the reviewers' approval.

Sincerely, 

Yong-Seok Lee, MD, PhD

Department of Neurology, College of Medicine Seoul National University, Seoul National University-Seoul Metropolitan Government Boramae Medical Center 

Dongjak-gu, Shindaebang-dong, Boramae 5 Road 20, 156-707, Seoul, Republic of Korea

Tel: 82-2-870-2476, Fax: 82-2-831-2826, E-mail: mercades@snu.ac.kr

And 

Hyun-Woo Nam, MD, PhD

Department of Neurology, College of Medicine Seoul National University, Seoul National University-Seoul Metropolitan Government Boramae Medical Center 

Dongjak-gu, Shindaebang-dong, Boramae 5 Road 20, 156-707, Seoul, Republic of Korea

Tel: 82-2-870-2471, Fax: 82-2-831-2826, E-mail: hwnam85@gmail.com

---

## [Editor Report · Decision Letter 1]

20 Oct 2021

Effect of obstructive sleep apnea on cerebrovascular compliance and cerebral small vessel disease

PONE-D-21-27488R1

Dear Dr. Lee,

We’re pleased to inform you that your manuscript has been judged scientifically suitable for publication and will be formally accepted for publication once it meets all outstanding technical requirements.

Kind regards,

Academic Editor

PLOS ONE

Additional Editor Comments (optional):

interesting and well-structured paper, I find it useful to clarify some aspects related to the complications of OSA. the authors improved the text in accordance with the reviewers' suggestions. Well done. Compliments
---

## [Editor Report · Acceptance letter]

3 Nov 2021

PONE-D-21-27488R1 

Effect of obstructive sleep apnea on cerebrovascular compliance and cerebral small vessel disease 

Dear Dr. Lee:

I'm pleased to inform you that your manuscript has been deemed suitable for publication in PLOS ONE. Congratulations! Your manuscript is now with our production department. 

Kind regards, 

on behalf of

Dr. Giannicola Iannella 

Academic Editor

PLOS ONE